# Exploring *Aeromonas veronii* in Migratory Mute Swans (*Cygnus olor*): A Debut Report and Genetic Characterization

**DOI:** 10.3390/vetsci12020164

**Published:** 2025-02-13

**Authors:** Zhifeng Peng, Chunyan Gao, Hongxing Qiao, Han Zhang, Huimin Huang, Yamin Sheng, Xiaojie Zhang, Baojun Li, Baoliang Chao, Jingjing Kang, Chuanzhou Bian

**Affiliations:** 1College of Veterinary Medicine, Henan University of Animal Husbandry and Economy, Zhengzhou 450046, China; zfp2017@hnuahe.edu.cn (Z.P.); 19712911505@163.com (C.G.); 80414@hnuahe.edu.cn (H.Q.); 221017@hnuahe.edu.cn (H.H.); 241048@hnuahe.edu.cn (Y.S.); 231045@hnuahe.edu.cn (X.Z.); 13783990302@163.com (B.C.); 2College of Veterinary Medicine, Henan Agricultural University, Zhengzhou 450046, China; zhanghan.zz@hotmail.com; 3Zhengzhou Zoo, Zhengzhou 450000, China; lijunbao.112233@163.com

**Keywords:** mute swan, *Aeromonas veronii*, emerging pathogen, zoonotic, public health

## Abstract

*Aeromonas veronii* (*A. veronii*) can cause serious disease in humans and various animals and is ubiquitous in terrestrial and aquatic milieus. Therefore, *A. veronii* has recently been considered to be an emerging pathogen worldwide. However, there is no documentation of clinical symptoms and pathological changes in *A. veronii* causing disease in migratory mute swans. In this study, we isolated and identified an *A. veronii* strain from migratory mute swans in China for the first time. We further explored its pathogenicity and antimicrobial susceptibility. Our results indicated that the newly identified strain has a strong ability to cause disease and exhibits multi-drug resistance. This is the first report of *A. veronii* being isolated from a migratory mute swan, which expands its known host spectrum. These findings not only help us to better understand the zoonotic potential of *A. veronii* but are also very significant in terms of improving the understanding of its dynamic transmission among the environment, animals, and humans.

## 1. Introduction

The 36 species of the genus *Aeromonas* are Gram-negative, facultatively anaerobic, non-spore-forming, and catalase-positive bacilli, which are widely distributed in municipal and hospital wastewater [1,2], lake water [3], river water, oceanic water, irrigation water, and regenerated wastewater [4,5]. While some species are opportunistic pathogens of humans [6,7], they also affect a variety of animals, including reptiles, amphibians, fish, and aquatic animals [8,9]. *Aeromonas* species can cause hepatobiliary tract, urinary tract, soft tissue, and skin infections, as well as pneumonias, peritonitis, and severe septicemia [6,7,9]. The pathogenic potential of *Aeromonas* is associated with several virulence factors, including aerolysin (*aerA*), hemolysin A (*hlyA*), heat-stable cytotonic toxins (*ast*), heat-labile cytotonic enterotoxin (*alt*), and aerolysin-related cytotoxic enterotoxin (*act*) [10,11]. In addition, the lipase (*lip*), lateral elastase (*ela*), polar flagellum (*fla*), flagella (*laf*), and type III secretion system (*TTSS*) contribute to the virulence of the genus *Aeromonas* [12,13]. The above-mentioned virulence factors allow *Aeromonas* bacteria to adhere, invade, and escape the host’s immune response [14,15].

As an important species of the genus *Aermonas*, *A. veronii* is pathogenic to various aquatic animals and terrestrial species. This bacterium has been isolated from sheep exhibiting respiratory symptoms [16], dead Yangtze finless porpoise in China [17], *Macrobrachium nipponense* with red gill disease in China [18], diseased *Trionyx sinensis* in China [19], diseased *Lateolabrax maculatus* in China [20], various freshwater fish species in Malaysia [21] and Thailand [22], marketed *Tegillarca granosa* in Korea [23], channel catfish in the United States [24], and farmed *Channa micropeltes* in Thailand [25]. More importantly, *A. veronii* infections have caused serious disease in humans, particularly in the elderly and children, causing sepsis, gastroenteritis, and other conditions [26,27]. It is a dominant bacterial pathogen in freshwater fish worldwide [21,28], and *A. veronii* is the most prevalent *Aeromonas* species within the genus *Aeromonas* in food products in Shanghai, China [29]. Therefore, *A. veronii* has become a dominant pathogenic species within the genus *Aermonas* in recent years. Additionally, humans and animals are mainly infected with *A. veronii* through the ingestion of contaminated water or food, exposure to fecal contamination, and direct contact [30,31]. All of the aforementioned studies highlight that *A. veronii* is involved not only in animal health and environmental health, but also human health and food safety, posing a serious threat to the One Health approach [27,28,31,32]. Consequently, *A. veronii* has been considered an emerging zoonotic bacterial pathogen worldwide [32].

The migratory mute swan is the most abundant of the swan species, with a total of seven populations distributed worldwide, comprising over 600,000 individuals [33]. They help regulate vegetation growth by foraging on aquatic vegetation, which affects water quality and provides habitats for other species. In addition, their aggressive territorial behavior has led endangered species like terns and black predators to abandon their nests [34]. In East Asia, the migratory mute swan migrates in a timed and directional pattern, stopping at specific stopover sites along three flyways among China, Russia, Mongolia, North Korea, South Korea, and Japan [35]. The presence of multi-drug-resistant zoonotic bacteria of various genus and different kinds of viruses in migratory mute swans enables them to potentially transmit zoonotic pathogens and drug-resistance genes over long distances, thus posing a potential threat to public health [35,36,37].

The prevalence of *Aeromonas* spp. in several migratory birds in China has been reported [36]. However, there is no documentation of clinical symptoms and pathological changes in *A. veronii* causing disease in migratory mute swans (*Cygnus olor*). In this study, *A. veronii* was first identified as causative agent of multi-organ lesions in migratory mute swan that died in China. Its pathogenicity and antimicrobial resistance were characterized, respectively. Our findings expand the list of hosts for *A. veronii* to include migratory mute swans, and the possible transmission routes and public health significance of *A. veronii* are also discussed.

## 2. Materials and Methods

### 2.1. Case Presentation and Pathological Examination

Many mute swans and various species of water birds inhabit Sanmenxia Swan Lake National Urban Wetland Park in the city of Sanmenxia, Henan province, China (Figure 1). In April 2022, several mute swans died successively without obvious clinical symptoms. A fresh corpse of mute swan was transported to the Veterinary Diagnostic Laboratory, Henan University of Animal Husbandry and Economy, for post-mortem examination. The mute swan weighed 12.8 kg and showed no wounds on the body surface (Figure 2A). After disinfection of the mute swan with 75% ethanol, a post-mortem examination was conducted, and various internal organs (spleen, lung, liver, heart, and kidney) were sampled aseptically for laboratory examinations. Tissues fixed with neutral 10% formalin were embedded in paraffin and then manually sectioned with a microtome to obtain 4–5 μm thick paraffin sections. Then, dewaxed sections were stained with hematoxylin and eosin (HE) for light microscopy. This study was approved by the Research Ethics Committee of Henan University of Animal Husbandry and Economy (approval number HNUAHE ER2324121).

### 2.2. Bacteria Culture and Molecular Detection of Virus

The visceral organs were streaked onto brain–heart infusion (BHI) plates with 5% sheep blood (AOBOX, Beijing, China), blood agar plates with 5% sheep blood (Biocell, Zhengzhou, China), MacConkey (MAC) plates (AOBOX, Beijing, China), and Trypticase soy agar (TSA) plates with 5% sheep blood (AOBOX, Beijing, China), respectively. After incubation at 37 °C for 24 h, single-cultured colonies were selected for bacteriological culture, Gram staining, and microscopic examination [38]. Then, the tissue samples were pooled and homogenized in phosphate-buffered saline (PBS, pH 7.2). After being freeze–thawed three times, the homogenized tissue samples were centrifuged at 8000× *g* for 10 min. Viral DNA/RNA was extracted from the supernatant using a Takara Virus DNA/RNA Kit (Takara, Dalian, China) according to the manufacturer’s instructions. The DNA/RNA samples were used to detect goose reovirus (GRV), goose hemorrhagic polyomavirus (GHPV), goose parvovirus (GPV), Tembusu virus (TMUV) and avian influenza virus (AIV) with specific primers as described previously [39,40,41] (Appendix A).

### 2.3. Protein Analysis by MALDI-TOF MS

According to the manufacturer’s instructions, the single-cultured colonies were chosen for species identification using MALDI-TOF MS (Bruker, Billerica, MA, USA) [42]. Approximately 1 μL of each extract was prepared and mixed with α-cyano-4-hydroxycinnamic acid (Bruker Daltonic, Billerica, MA, USA), spotted onto a 96 polished steel targets plate, and dried at room temperature. Mass spectrometry was performed with a MALDI Biotyper system with the microflex^®^LRF software V3.0 (Brucker Daltonics). Spectra ranging from a mass-to-charge ratio (*m*/*z*) of 2000 to 20,000 were determined using the MALDI Biotyper system the V4.1 software and library (MALDI Biotyper database, 8468 entries, Bruker Daltonics).

### 2.4. Sequence Analysis of 16S rRNA and gyrB Genes

The genomic DNA of the newly identified strain was extracted using a genomic DNA purification kit (Tiangen Biotech, Beijing, China) following the manufacturer’s guidelines and then stored at −20 °C. The *16S rRNA* was amplified using universal primers 27F (5′-AGAGTTTGATCCTGGCTCAG-3′) and 1492R (5′-GGCTACCTTGTTACGACTT-3′) and a PCR amplification procedure as described previously [43]. The *gyrB* gene was amplified using the universal primers gyrB-F (5′-GAAGTCATC ATGACCGTTCTGCAYGCNGGNGGNAARTTYGA-3′) and gyrB-R (5′-AGCAGGGTACGGATGTGCGAGCCRTCNACRTCNGCRTCNGTCAT-3′) and a PCR amplification procedure as described previously [44]. The *16S rRNA* and *gyrB* gene sequences were sequenced by SUNYA Biotech (Zhengzhou, China). Further molecular identification of the isolated strains was performed through phylogenetic analysis of the *16S rRNA* and *gyrB* genes. Phylogenetic trees were constructed based on *16S rRNA* gene and *gyrB* gene sequences and analyzed using the MEGA-X software with the neighbor-joining method and 1000 bootstrap replicates, respectively.

### 2.5. Detection of Virulence-Associated Genes

Total DNA extracted from the newly identified *Aeromonas* isolate was used as PCR templates for virulence gene detection. Virulence genes *aerA*, *hlyA*, *fla*, *alt*, *ast*, *act*, *lip*, and *ela* were selected as virulence markers. The virulence genes were detected using the primers (Appendix A) and PCR amplification procedure as described previously [38]. Analysis of PCR products by 1% agarose gel electrophoresis was performed using a DL 2000 DNA marker (Takara, Dalian, China).

### 2.6. Pathogenicity Test on Goslings

Twenty-five healthy, one-day-old goslings were obtained from Yuanyang Yunbo goose industry company. Goose reovirus (GRV), goose hemorrhagic polyomavirus (GHPV), goose parvovirus (GPV), Tembusu virus (TMUV) and avian influenza virus (AIV) were excluded using specific primers as described previously [40,41]. To acclimate the laboratory environment, all goslings were housed in a pathogen-free environment for three days before the experiment initiation. Four groups of goslings (five goslings per group) were inoculated intraperitoneally with the newly identified strain at doses of 1.1 × 10^9^, 1.1 × 10^8^, 1.1 × 10^7^, and 1.1 × 10^6^ CFU/gosling (0.2 mL/gosling), respectively. One control group was inoculated with 0.2 mL sterile PBS per gosling. The mortality and symptoms of the goslings were recorded daily for 7 days. The LD_50_ value was calculated by the Reed and Muench method. Three dead goslings were randomly selected for bacteriological detection and pathological examination. The soft tissues fixed with neutral 10% formalin were embedded in paraffin and then manually sectioned with a microtome to obtain 4–5 μm thick paraffin sections. Then, dewaxed sections were stained with HE for light microscopy. Re-isolation and identification of the bacteria from the hearts, livers, and spleens of dead goslings were carried out.

### 2.7. Antimicrobial Susceptibility Testing

The antimicrobial susceptibility was evaluated using the microbroth dilution method according to the Clinical and Laboratory Standards Institute guidelines (CLSI, 2020). Briefly, a single colony of the newly identified *Aeromonas* isolate was inoculated onto Mueller–Hinton (MH) agar and incubated at 35 ± 1 °C for 24 h. A 0.5 McFarland turbidity standard (5 × 10^5^ CFU/mL) bacterial suspension was prepared with MH broth. Then, 50 µL of the bacterial suspension was added to each well of the microplate. Two wells containing only bacterial suspension were used as a positive control, and two wells containing only sterile cation-adjusted MH broth were used as a negative control. The microplates were incubated at 35 ± 1 °C for 24 ± 2 h. *Escherichia coli* ATCC 25922 was used as a quality control strain in all tests. The following antibiotics (Solarbio, Beijing, China) were tested: meropenem (0.25–128 µg/mL), linezolid (0.25–128 µg/mL), ampicillin (0.25–128 µg/mL), doxycycline (0.0625–32 µg/mL), tigecycline (0.25–128 µg/mL), colistin (0.25–128 µg/mL), gentamicin (0.25–128 µg/mL), florfenicol (0.125–64 µg/mL), enrofloxacin (0.3125–160 µg/mL), cefoxitin (0.25–128 µg/mL), ceftazidime (0.25–128 µg/mL), tiamulin (1–512 µg/mL), spectinomycin (0.25–128 µg/mL), and Fosfomycin (0.25–128 µg/mL).

## 3. Results

### 3.1. The Dead Mute Swan Exhibited Acute Lesions in Multiple Organs

After necropsy, multiple organs of the dead mute swan exhibited acute lesions, including degeneration and hemorrhage in the myocardium (Figure 2B), swelling in the liver (Figure 2C), hemorrhage in the spleen (Figure 2D), hemorrhage and swelling in the kidneys (Figure 2E), and congestion and necrosis in the lungs (Figure 2F). Based on hemorrhage in multiple organs, the cause of death of the mute swan was initially diagnosed as bacteremia and septicemia. The histological examination of the dead mute swan showed inflammatory cell infiltration, vacuolar degeneration, and extensive hemorrhage in multiple organs. The myocardial extracellular spaces were widened, with a large number of inflammatory cells infiltrating (Figure 2G). The structure of hepatic cords was unclear, and local necrosis was infiltrated by inflammatory cells (Figure 2H). The spleen exhibited capillary congestion (Figure 2I). The alveolar interstitium was widened, with a large number of inflammatory cells infiltrating, and a great quantity of red blood cells was found in the alveolar interstitium (Figure 2J).

### 3.2. The Isolated Strain HNZZ-1/2022 Was a Gram-Negative, Short Rod with Blunt Ends

The aseptically collected tissues were inoculated on TSA plates, MAC plates, BHI plates, and sheep blood plates, respectively, for bacterial isolation. The results showed that the dominate bacteria, termed HNZZ-1/2022, was cultured from plates inoculated with the heart blood, spleen, lung, and liver tissue. The colonies of HNZZ-1/2022 were observed to be smooth, round, and pale grey in color, with a diameter of 2 mm (Figure 3A). The newly isolated strain HNZZ-1/2022 exhibited β-hemolysis on sheep blood plates (Figure 3B), and it was Gram-negative, arranged singly or doubly, and was a short rod with blunt ends (Figure 3C).

### 3.3. The Strain HNZZ-1/2022 Was Identified as A. veronii

In the protein analysis by MALDI-TOF MS, the strain HNZZ-1/2022 best matched with *A. veronii* with a score value of 2.255. The *16S rRNA* gene of the isolated bacterial strain HNZZ-1/2022 was 1459 bp in length (Appendix A; GenBank accession number: OQ860819). It was 99.6–100.0% identical to *A. veronii* ATCC 35624 (NR_118947.1), *A. veronii* AD-AV201606 (MG736237.1), and *A. veronii* JCM 7375 (NR_112838.1). The *gyrB* gene was 1191 bp in length (Appendix A; GenBank accession number: PP798204), and it was 98.3–99.4% identical to *A. veronii* LOID15995 (CP121859.1) and *A. veronii* Colony111 (CP070207.1). In the phylogenetic trees constructed based on the *16S rRNA* sequence (Figure 3D) and *gyrB* sequence (Figure 3E), the isolated strain HNZZ-1/2022 was clearly grouped with a cluster of known *A. veronii* strains. Based on the morphology, MALDI-TOF testing, and analysis of the *16S rRNA* and *gyrB* gene sequences, the isolated strain HNZZ-1/2022 was identified as *A. veronii*.

### 3.4. Strain HNZZ-1/2022 Contained Six Virulence-Associated Genes

To evaluate the virulence of *A. veronii* strain HNZZ-1/2022, eight virulence-associated genes were detected by PCR. The results indicated that *A. veronii* HNZZ-1/2022 contained six virulence-associated genes, including genes encoding for heat-labile cytotoxic enterotoxin (*alt*), elastase (*ela*), lipase (*lip*), cytotoxic enterotoxin (*act*), aerolysin (*aerA*), and polar flagella (*fla*) (Figure 4).

### 3.5. A. veronii Strain HNZZ-1/2022 Caused Multiple-Organ Pathological Changes in Goslings

Four groups of goslings (five goslings per group) were intraperitoneally inoculated with the newly identified strain with doses of 1.1 × 10^9^, 1.1 × 10^8^, 1.1 × 10^7^, and 1.1 × 10^6^ CFU/gosling (0.2 mL/gosling), respectively. The goslings exhibited various symptoms, such as lethargy, tremor, and tachypnea, within 12 h post inoculation, which were similar to those in the dead mute swan. The goslings began to die 24 h post inoculation, and all the goslings in the two highest-dose inoculation groups died within 2 days. The mortality rates of the other three groups inoculated with doses of 1.1 × 10^7^ and 1.1 × 10^6^ CFU per gosling within 7 days were 0% and 0%, respectively. The LD_50_ value of *A. veronii* strain HNZZ-1/2022 was estimated to be 3.48 × 10^8^ CFU/mL for goslings (Table 1). No obvious pathological lesions were observed in the negative control group (Figure 5A,D). Pathological lesions in the heart, liver, and lung were similar to those in the dead mute swan (Figure 5B,C,E–G). Additionally, according to the sequence analysis of the *16S rRNA* gene and *gyrB* gene, the recovered isolate from the infected goslings was identified as *A. veronii*.

### 3.6. Strain HNZZ-1/2022 Is Multi-Drug-Resistant

The antibiotic susceptibility test of the *A. veronii* HNZZ-1/2022 strain using the microbroth dilution method using 14 antibiotics showed that HNZZ-1/2022 was sensitive to cefoxitin (8 µg/mL) and gentamicin (2.5 µg/mL) and resistant to meropenem (32 µg/mL), ampicillin (32 µg/mL), and enrofloxacin (0.625 µg/mL) (Appendix A).

## 4. Discussion

To date, *A. veronii* has been reported as a causative pathogen in various species of animals, including freshwater fish, amphibians, birds, aquatic animals, and reptiles [8,9,45,46]. *A. veronii* has caused various diseases in humans worldwide, and it has become the dominant *Aeromonas* species among clinical isolates [21,24,28,47,48,49]. Additionally, it is involved in food safety and public health concerns [28,29,32]. Given these facts, *A. veronii* has become an increasingly recognized zoonotic pathogen due to its isolation from various clinical and environmental samples worldwide [3,32,50]. In the present study, *A. veronii* was isolated as the causative agent from a dead migratory mute swan exhibiting acute lesions in multiple organs, such as hemorrhage and necrosis. To our knowledge, this is the first report of a fatal *A. veronii* infection in migratory mute swans, thus expanding the known host spectrum of *A. veronii*, which was traditionally limited to aquatic animals and humans [12,26,27].

Previous studies have shown that the virulence phenotype is a cumulative effect of multiple pathogenic factors [51]. In this study, *A. veronii* HNZZ-1/2022 contained six virulence genes, including genes encoding for heat-labile cytotoxic enterotoxin (*alt*), elastase (*ela*), lipase (*lip*), cytotoxic enterotoxin (*act*), aerolysin (*aerA*), and polar flagella (*fla*). These virulence genes encode proteins and are considered to be major factors affecting the pathogenicity of the genus *Aeromonas*. The presence of polar flagella (*fla*) in the genus *Aeromonas* confers rapid mobility, allowing the bacteria move on solid surfaces and form biofilms [10]. Additionally, the cytotoxic enterotoxin (*act*) restrains the phagocytosis activity of host cells and produces hemolysis [52]. The aerolysin encoded by the *aerA* gene is the prototype hemolysin of the genus *Aeromonas*, which can form pores in the target cell membrane and lead to osmotic lysis [10]. In this study, the LD_50_ value of the *A. veronii* strain HNZZ-1/2022 was estimated to be 3.48 × 10^8^ CFU/mL for goslings. These findings indicate that HNZZ-1/2022 is highly pathogenic.

With the increasing reports of multi-drug-resistant *A. veronii*, antimicrobial resistance has become an increasingly concern in humans as well as animals [24,29,50]. *Aeromonas* species isolated from aquatic organisms, insects, chicken, and humans in India show similar antimicrobial resistance profiles [53]. Previous studies indicated that most *A. veronii* strains are resistant to amoxicillin, ampicillin, and penicillin, regardless of whether isolated from humans or animals [8,54]. In the current study, *A. veronii* HNZZ-1/2022 isolated from migratory mute swan exhibited resistance to meropenem, ampicillin, and enrofloxacin. Carbapenems are considered critically important antimicrobials for human medicine by the World Health Organization. Notably, carbapenemase-producing *A. veronii* has been reported in the environment of an equine veterinary hospital in the USA [50]. Additionally, recent studies indicated that *A. veronii* strains isolated from the urban-impacted Akaki river in Ethiopia were resistance to imipenem (75.5%) and meropenem (63.3%), with 94.4% of carbapenem-resistant *A. veronii* strains being carbapenemase producers [55]. Furthermore, almost all *A. veroni* strains isolated from municipal and untreated hospital wastewater were also resistant to imipenem and meropenem [2]. In the current study, the newly identified *A. veronii* HNZZ-1/2022 isolated from migratory mute swans exhibited resistance to meropenem. The emergence of bacterial resistance to carbapenem antibiotics poses a significant challenge to public health, as carbapenems are a last-resort treatment option for life-threatening infections. Resistance to these drugs would reduce the effectiveness of infection treatment and increase medical costs. Moreover, *A. veronii* HNZZ-1/2022 showed penicillin resistance, which is consistent with previous reports that Aeromonads are intrinsically resistant to β-lactams [22].

Although we cannot confirm the sources of the bacteria in this case, it is possible that *A. veronii* was horizontally transmitted through contaminated water or food. There is a possibility that the mute swans were exposed to the bacteria via contaminated water or fish raised in the lake. Additionally, tourists sometimes feed mute swan sweet potatoes and other foods, which may also serve as a source of infection. Alternatively, the mute swan may have been infected through ingestion of food contaminated with *A. veronii*. The data obtained in the current study suggest that the migratory mute swan is a natural reservoir of *A. veronii*, implying mute swans can transmit the pathogen over long distances. In East Asia, migratory mute swans migrate along three routes connecting China, Russia, Mongolia, North Korea, South Korea, and Japan. During migration, mute swans stop at various parks, rivers, and freshwater lakes such as Qinghai Lake, Valley of the Lakes, Hulun Lake, Yellow River, Selenga River, Har Us Nuur National Park, and Khogno Khan National Park [35]. These findings suggest that *A. veronii* can be transmitted to many areas containing bodies of water by migratory mute swans, leading to water contamination [36]. Humans can be infected with *A. veronii* through direct and indirect contact with the migratory mute swans or through drinking *A. veronii*-contaminated water. Meanwhile, fish obtained from contaminated water can cause foodborne transmission of *A. veronii* in humans [29,32]. Furthermore, farmed waterfowl can be directly and indirectly infected with carbapenem-resistant *A. veronii* through drinking contaminated water, potentially transmitting the bacteria to humans. Therefore, foodborne and waterborne transmission of *A. veronii* is a public health concern that should receive more attention. This calls for continuous monitoring of *A. veronii* from various sources for One Health mitigation as well as improved environmental sanitation for farmed waterfowl to prevent contamination of water and human infections. It also highlights the need for caution when using carbapenems in the clinical treatment of *A. veronii* infections.

## 5. Conclusions

To the best of our knowledge, this is the first report of *A. veronii* being isolated from a migratory mute swan in China, further expanding its known host spectrum. A gosling pathogenicity test showed that *A. veronii* HNZZ-1/2022 could cause goslings to die. Moreover, it exhibits resistance to meropenem, ampicillin, and enrofloxacin. These findings suggest its strong zoonotic potential. However, in the current study, only one *A. veronii* strain isolated from a migratory mute swan in Sanmenxia Swan Lake National Urban Wetland Park was studied, and the mechanism of its antibiotic resistance was not studied deeply. Hence, much more effort will be focused on the active surveillance of *A. veronii*, its antibiotic resistance, and its transmission mechanism to better understand the epidemiological characteristics of *A. veronii* and guide the management of future human and animal infections.

## Figures and Tables

**Figure 1 vetsci-12-00164-f001:**
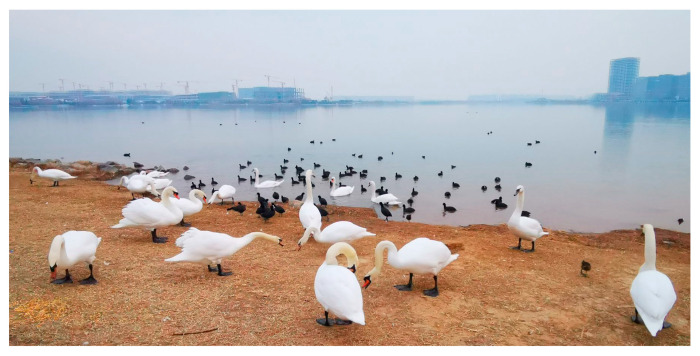
Migratory mute swans (*Cygnus olor*) at the Sanmenxia Swan Lake National Urban Wetland Park in Henan, China. Migratory whooper swans feed on the shore and in the water at the Sanmenxia Swan Lake National Urban Wetland Park in Henan province, China.

**Figure 2 vetsci-12-00164-f002:**
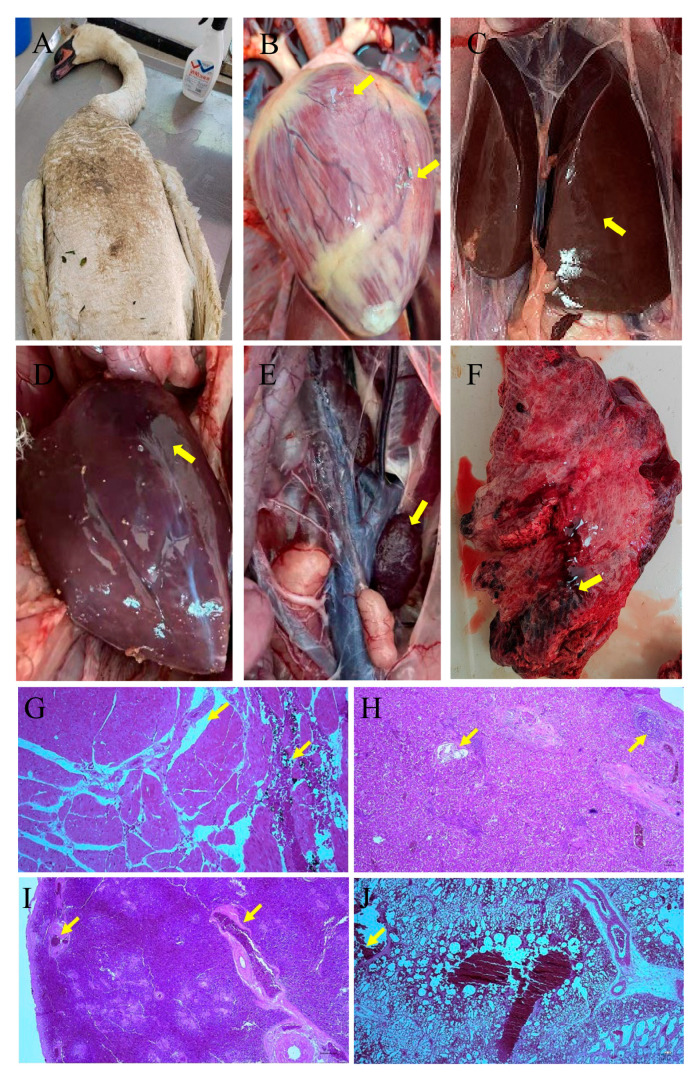
Gross and histological lesions of the dead migratory mute swan in Henan province, China. HE. (**A**) Emaciation without visible wounds in the dead mute swan; (**B**) degeneration and hemorrhage in the myocardium; (**C**) swelling in the greyish-yellow liver; (**D**) edema and congestion in the spleen; (**E**) renal swelling; (**F**) congestion and necrosis in the lung; (**G**) the myocardial extracellular spaces were widened, and there was a large amount of inflammatory cell infiltration (100×); (**H**) the structure of the hepatic cord was unclear, and local necrosis was infiltrated by inflammatory cells (100×); (**I**) the spleen exhibited capillary congestion (40×); (**J**) alveolar interstitial space was widened with a large amount of inflammatory cell infiltration. A large number of red blood cells were found in the alveolar cavity and alveolar interstitium (100×).

**Figure 3 vetsci-12-00164-f003:**
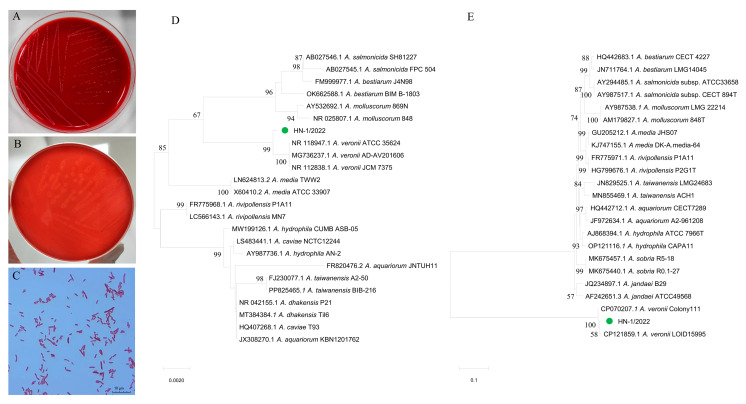
Isolation and identification of bacterial strain HNZZ-1/2022. (**A**,**B**) The bacteria grew well and produced β hemolysis band on sheep blood agar plates; (**C**) Gram-negative bacilli with varying sizes and staining depths were observed under oil immersion. Phylogenetic trees based on *16S rRNA* gene (**D**) and *gyrB* gene (**E**). The phylogenetic trees were constructed using the neighbor-joining method with 1000 bootstrap replicates in the MEGA-X software (MEGA_X_CC_10.2.6). Green circles represent the newly isolated *A. veronii* strain in the present study.

**Figure 4 vetsci-12-00164-f004:**
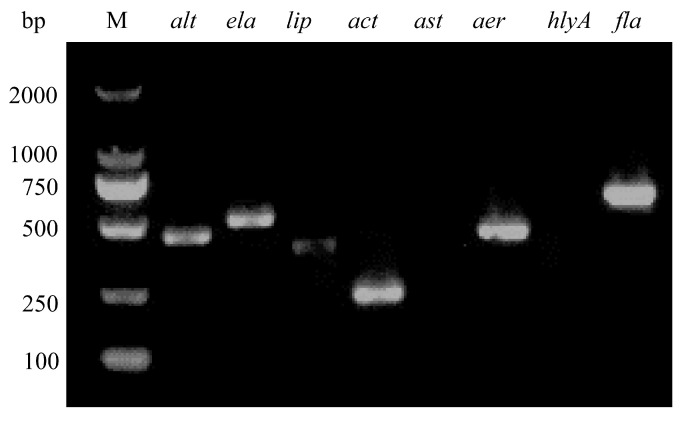
PCR results of virulence genes in *A. veronii* HNZZ-1/2022. Lane M: DL2, 000 DNA marker. Six virulent genes (*alt*, *ela*, *lip*, *act*, *aer*, and *fla*) were present in HNZZ-1/2022.

**Figure 5 vetsci-12-00164-f005:**
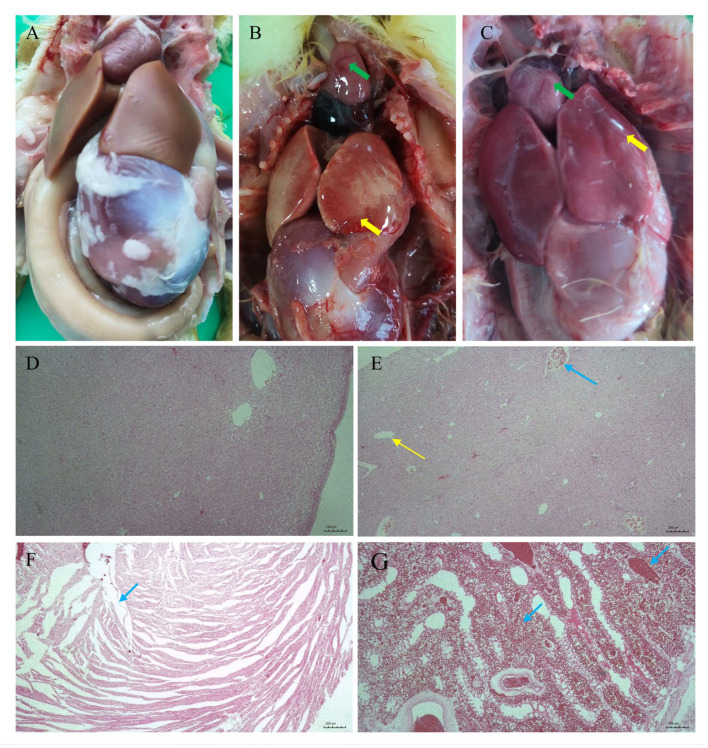
The pathological changes in and histological lesions of the goslings infected with *A. veronii* HNZZ-1/2022. HE. (**A**) No visible lesions in the negative control goslings; (**B**,**C**) degeneration and hemorrhage in the heart (green arrow); (**B**,**C**) swelling and hemorrhage in the liver (yellow arrow); (**D**) liver_NC (40×); (**E**) severe intravascular congestion (blue arrow) and severe swelling of liver cells, karyolysis, and cell necrosis (yellow arrow) (40×); (**F**) fatty degeneration of myocardium (40×); (**G**) alveolar space and alveolar interstitial space had a large number of red blood cells (40×). Bar = 200 µm.

**Table 1 vetsci-12-00164-t001:** Determination of median lethal dosage (LD_50_) of *A. veronii* HNZZ-1/2022.

Group	Dose (CFU/Gosling)	Death Number (5 Gosling/Group)	Cumulative Mortality	LD_50_ (CFU/mL)
1	1.1 × 10^9^	4	83.33%	3.48 × 10^8^
2	1.1 × 10^8^	1	16.67%	
3	1.1 × 10^7^	0	0%	
4	1.1 × 10^6^	0	0%	

## Data Availability

All the data generated and analyzed in this study are included in the published article and its Appendix A.

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
