# Peer review of "Exploring *Aeromonas veronii* in Migratory Mute Swans (*Cygnus olor*): A Debut Report and Genetic Characterization"

_vetsci, 2025, doi:10.3390/vetsci12020164_

Round 1

Reviewer 1 Report

Comments and Suggestions for Authors

The authors present a priority finding that the newly isolated A. veronii had the ability to cause migratory swan disease, which was confirmed by experimental infection of goslings.

I have only a few minor comments about the manuscript:

l. 89: paragraph 2.1. Case presentation and pathological examination: This paragraph must also describe the histological examination procedure, including staining techniques. The staining technique should also be mentioned in Figure 2, including magnification.

l. 106 : “with 5% erythrocytes” change to  “with 5% sheep blood”

l. 179-181:   the source of the substances and the range of tested concentrations for each substance must be added

l. 253, 295, Table 1: An error may have occurred while calculating the LD50 value.

Author Response

Please see the attachment. Thank you so much for your constructive and kindly comments, which is highly significant to us.

Reviewer 2 Report

Comments and Suggestions for Authors

The work entitled Exploring Aeromonas veronii in migratory mute swans (Cygnus olor): A Debut Report and Genetic Characterization” makes a methodical description of the postmortem description of mute swans in China, in which the damage caused by the infection of the Aeromonas veronii bacteria is studied histopathologically, which they identify by sequencing of 16S rRNA and gyrB using generic primers.

The exhaustive study uses traditional microbiology methods, massive sequencing, and MALDI-TOF MS analysis to determine the pathogen, confirmed by infection tests. For the first time, it describes the clinical picture of a mute swan infected with Aeromonas veronii.

The work is well done and detailed, and experimentation consolidates observations.

It is acceptable, in its current state, for publication.

Author Response

Please see the attachment. Thank you so much for your kindly comments.

Reviewer 3 Report

Comments and Suggestions for Authors

Dear author,

Thank you for your work

My comments to the manuscript as below:

Title: accepted as can get attention from reader

Abstract: accepted. But please provide full scientific name for the first mention

If possible provide value for the mentioned parameters

Introduction

may provide the importance of the mute swans (Cygnus olor) in ecological aspect in a paragraph

Materials and methods

Provide ethical statement in the 2.1

any references for the method in the lines 105-110?

L 178-181: provide brand and country maker of the mentioned antibiotics

Discussion

I think author is missing to discuss in detail the findings of antibiotic results

Conclusion

Need to put more efforts by providing research gap and future work

furthermore, providing summary of each finding of the study will benefit the section

Author Response

(The authors gave the same response as above.)
